# Uncertainty Quantification for Inferring Hawkes Networks

**Haoyun Wang[1], Liyan Xie[1], Alex Cuozzo[2], Simon Mak[2], Yao Xie[1]**
[1]Georgia Institute of Technology, [2] Duke University

## Abstract

Multivariate Hawkes processes are commonly used to model streaming networked event data in a wide variety of applications. However, it remains a challenge to extract reliable inference from complex datasets with uncertainty quantification. Aiming towards this, we develop a statistical inference framework to learn causal relationships between nodes from networked data, where the underlying directed graph implies Granger causality. We provide uncertainty quantification for the maximum likelihood estimate of the network multivariate Hawkes process by providing a non-asymptotic confidence set. The main technique is based on the concentration inequalities of continuous-time martingales. We compare our method to the previously-derived asymptotic Hawkes process confidence interval, and demonstrate the strengths of our method in an application to neuronal connectivity reconstruction.

## 1   Introduction

Recently, there has been a surge of interest in using Hawkes processes networks to model discrete events data in both the statistics and the machine learning community (see a review in [20]). The popularity of the model can be attributed to its wide range of applications, including seismology, criminology, epidemiology [24], social networks [14], neural activity [22], and so on. The model is capable of capturing a spatio-temporal triggering effect reflected in real-world networks – one event may trigger subsequent events at different locations. Existing works for recovery of the Hawkes network focus on performing point estimators: most of them rely on estimating influence coefficients (representing the magnitude of the influence) and thresholding by a pre-specified value to recover the Hawkes network structure. However, most existing work does not quantify the uncertainty, for instance, in the form of a confidence interval.

One outstanding issue with point estimators is that, without accurate uncertainty quantification, one cannot claim any statistical significance of the results. For instance, it is difficult to assign a probability to the existence of a directed edge between two nodes. This problem is critical for certain problems, such as causal inference. In Hawkes network models, Granger causality has a very simple form: there is a causal relationship between two nodes in the network if and only if there exists an edge between the two nodes, and there is no casual relationship otherwise [7]. Thus, uncertainty quantification (UQ) is critical in scientific studies, because we wish to test whether a causal relationship exists between one node to another node at a given statistical confidence level. The development of easy-to-implement and robust UQ tools is crucial for a variety of scientific problems, from medical data to social networks to neural spike train data and more.

In this paper, we study the uncertainty quantification for maximum likelihood estimates of multivariate Hawkes processes over networks. Each node can represent a location, a region of a brain, or a user in a social network. We are particularly interested in recovering the underlying structure (connections) between different nodes with uncertainty quantification, meaning providing accurate upper and lower confidence intervals for the influence coefficients (which is linked to the causal relationships between

these nodes). Since we are particularly interested in the existence (or non-existence) of an underlying edge, one of the most critical aspects of our study is to perform network topology recovery. Motivated by applications where the recovery guarantee is usually required, we focus on quantifying the uncertainty of the maximum likelihood estimate of the unknown parameters by providing confidence intervals (CIs). We proposed a novel non-asymptotic approach to establish a general confidence polyhedral set for hidden network influence parameters (from which the confidence sets can be extracted). The non-asymptotic confidence set is established by constructing a continuous-time martingale using the score function (the gradient of the log-likelihood function) of the network Hawkes process. This enables us to apply a concentration bound for continuous-time martingales. The non-asymptotic confident set is more accurate than the classic asymptotic confidence intervals, since the concentration bound approach captures more than the first and second-order moments (which are essentially what the asymptotic confidence intervals are capturing). We compared the two methods for establishing CIs using synthetic neural activity data, to demonstrate the effectiveness of our approach.

**Contributions.** Our main contribution can be summarized as follows: (1) We give a non-asymptotic confidence set for the maximum likelihood estimate (MLE) of the Hawkes process over networks, and (2) Our confidence set is more general and can be approximated by a polyhedron; the CIs can be solved efficiently from a linear program. In contrast, the classic CI essentially provides a box in the high-dimensional space for the multi-dimensional parameters.

**Related Work.** There has been much effort made on network inference for multivariate point processes. Learning algorithms for Granger causality of Hawkes processes has been proposed in [28] using the regularized MLE. [1] proposed a nonparametric way to estimate the mutual inference and causality relationship in multivariate Hawkes processes. [29] considers the spatiotemporal Hawkes process and develop a nonparametric method for network reconstruction. [15] studies the detection of changes in the underlying dynamics. Moreover, recent work has also focused on causal inference for different applications, such as online platforms [13], infectivity matrix estimation [28], etc.

However, there is relatively little literature that provides theoretical guarantees on the significance level of the estimation results. The concentration results for inhomogeneous Poisson processes were studied in [21]. The non-asymptotic tail estimates for the Hawkes process were established in [23]. [6] studies Granger causality for brain networks and characterizes the significance level using numerical methods, while our result gives a theoretical guarantee on the confidence level. The CI for parameter recovery of discrete-time Bernoulli processes is given in [11]. At the same time, this paper focuses on the continuous-time Hawkes process, which is more complicated in uncertainty quantification. In Bayesian statistics literature, works have been done in quantifying the uncertainty of the network parameters by imposing a prior model on the model hyperparameters; the posterior is approximated using Markov chain Monte Carlo [19, 26]. Recently, there has been an effort to establish time-uniform CI based on concentration inequalities [10, 9].

The field of uncertainty quantification itself is very broad, with important applications in computer simulations [25], aerospace engineering [16] and climatology [18]. This literature can be grouped into two categories [27]: inverse UQ (the inference of parameters from a generating model) and forward UQ (the propagation of uncertainty through numerical models). The current work focuses on the inverse problem for the network multivariate Hawkes process and, in particular, in providing the confidence interval for the maximum likelihood estimates of parameters.

## 2 Background

A temporal point process is a random process whose realization consists of a list of discrete events localized in time. Let $(u_1, t_1), \cdots, (u_n, t_n)$ be a series of events happened during time period $[0, T]$ on a multivariate Hawkes process with $D$ nodes, where $t_i$ denoted the time of the $i$-th event, and $u_i \in [D]$ is the index of node where the event happens. The intensity function at node $i$ at time $t$ is

$$\lambda_i(t) = \mu_i + \sum_{j : t_j < t} \alpha_{i, u_j} \varphi_{i, u_j}(t - t_j), \ i = 1, \cdots, D,$$

where $\mu_i$ is the background rate of events happening at $i$-th node, $\alpha_{ij} \geq 0$ is a parameter representing the influence of node $j$ to node $i$, and $\varphi_{ij}$ is a function supported on $[0, \infty)$. Let $\boldsymbol{N}_t \in \mathbb{N}^D$ be a vector where the $i$-th entry $N_t^i$ is the number of events happened on node $i$ during $[0, t)$. For any

function $f$, define the following integral with counting measure

$$\int_0^T f(t)dN_t = \sum_{t \in \mathcal{H}_T} f(t),$$

where $\mathcal{H}_t = \{t_1, \ldots, t_n : t_n < t\}$ denotes the list of times of history events up to but not including time $t$. Therefore, we can rewrite the intensity function as

$$\lambda_i(t) = \mu_i + \sum_{j=1}^{D} \int_0^t \alpha_{ij}\varphi_{ij}(t-\tau)dN_\tau^i, \ i = 1, \cdots, D. \tag{1}$$

We may consider different types of influence function $\varphi$, including: (i) the *gamma function* $\varphi(\Delta t) = (\Delta t)^{k-1}e^{-\Delta t/\beta}/(\Gamma(k)\beta^k)$, $\Delta t \geq 0$. Note that when $k = 1$, it becomes the commonly-used *exponential function*, $\varphi(\Delta t) = \beta e^{-\beta \Delta t}$, $\Delta t \geq 0$, which shows that the influence of events on future intensity is exponentially decaying. The decay starts immediately following the onset (thus, there is no delay); (ii) the *Gaussian function*: $\varphi(\Delta t) = \exp\{-\beta(\Delta t - \tau)^2/\sigma\}/\sqrt{2\pi\sigma}$, $\Delta t \geq 0$, where $\tau \geq 0$ is the unknown delay which means that the influence attains its maximum value $\tau$ time after the event happens.

## 2.1 Decoupled log-likelihood function

Let $A = (\alpha_{ij})_{i,j \in [D]}$, $\alpha_{ij} \geq 0$, be a matrix that contains all the influence parameters between nodes and our parameter-of-interest to be estimated. Given the events on $[0, T]$, the likelihood function is (detailed derivation can be found in, e.g., [20])

$$L(A) = \exp\left(-\sum_{i=1}^{D}\int_0^T \lambda_i(t)dt\right)\prod_{j=1}^{n}\lambda_{u_j}(t_j),$$

where $\lambda_i(t)$ is the intensity as defined in (1). Note that $\lambda_i(t)$ depends on $A$, but we omit the term for simplicity.

The log-likelihood function can be written in the form of integral with counting measure $N_t^i$ at $i$-th node,

$$\ell(A) = \log L(A) = \sum_{i=1}^{D}\left(-\int_0^T \lambda_i(t)dt + \int_0^T \log \lambda_i(t)dN_t^i\right).$$

We note that the log-likelihood function $\ell(A)$ can be decoupled into summation of $D$ terms, each for a specific node,

$$\ell(A) = \sum_{i=1}^{D}\ell_i(\boldsymbol{\alpha}_i),$$

where $\boldsymbol{\alpha}_i := [\alpha_{i1}, \cdots, \alpha_{iD}]^{\mathsf{T}} \in \mathbb{R}^D$ is a column vector denoting influence of other nodes to node $i$, and

$$\ell_i(\boldsymbol{\alpha}_i) = -\int_0^T \lambda_i(t)dt + \int_0^T \log(\lambda_i(t))dN_t^i. \tag{2}$$

Since $\ell_i(\boldsymbol{\alpha}_i)$ only depends on the parameter $\boldsymbol{\alpha}_i$, the statistical inference for each node (therefore each $\boldsymbol{\alpha}_i$) can be decoupled, which enables us to perform the computation in parallel and simplify our analysis. For the rest of this paper, we focus on the inference of a single $\boldsymbol{\alpha}_i$.

**Notation.** We use $\boldsymbol{\alpha}_i^*$ to denote the true parameter which is unknown, $\widehat{\boldsymbol{\alpha}}_i$ to denote the estimated parameter for $i$-th node, $\widehat{\lambda}_i(t)$ to denote the intensity computed using the estimator $\widehat{\boldsymbol{\alpha}}_i$, and $\lambda_i^*(t)$ to denote the intensity under the true parameter $\boldsymbol{\alpha}_i^*$.

## 2.2 Score function and Fisher information

The statistical properties of the multivariate Hawkes process are closely related to its score function and the Fisher Information. The score function for $i$-th node given data, is defined as as

$$S_i(\boldsymbol{\alpha}_i) = \frac{\partial\, \ell_i(\boldsymbol{\alpha}_i)}{\partial \boldsymbol{\alpha}_i} = -\int_0^T \frac{\partial \lambda_i(t)}{\partial \boldsymbol{\alpha}_i} dt + \int_0^T \lambda_i^{-1}(t)\frac{\partial \lambda_i(t)}{\partial \boldsymbol{\alpha}_i} dN_t^i$$

$$= \int_0^T \lambda_i^{-1}(t)\frac{\partial \lambda_i(t)}{\partial \boldsymbol{\alpha}_i}(dN_t^i - \lambda_i(t)dt), \tag{3}$$

where $\partial \lambda_i(t)/\partial \boldsymbol{\alpha}_i$ is a vector independent of the choice of $\boldsymbol{\alpha}_i$. For simplicity, we denote this gradient by $\boldsymbol{\eta}_i(t) \in \mathbb{R}^D$, with $j$-th entry being

$$\eta_{ij}(t) = \frac{\partial \lambda_i(t)}{\partial \alpha_{ij}} = \int_0^t \varphi_{ij}(t-\tau)dN_\tau^j. \tag{4}$$

Note that $\eta$ includes information about the influence function between two nodes; this holds for any general influence function $\varphi_{ij}$.

The $D$-by-$D$ Hessian matrix of the log-likelihood function, given observations $dN_t^i$, is then

$$H_i(\boldsymbol{\alpha}_i) = \frac{\partial^2 l_i(\boldsymbol{\alpha}_i)}{\partial \boldsymbol{\alpha}_i \partial \boldsymbol{\alpha}_i^{\mathsf{T}}} = -\int_0^T \lambda_i^{-2}(t)\boldsymbol{\eta}_i(t)\boldsymbol{\eta}_i^{\mathsf{T}}(t)dN_t^i. \tag{5}$$

The Fisher Information $I_i^*$ is defined as the expected variance of the score function $S_i(\boldsymbol{\alpha}_i)$, and also as the negative expected Hessian of the log-likelihood function over unit time, assuming the process is stationary under true parameter $\{\boldsymbol{\alpha}_i^*\}_{i=1}^D$,

$$I_i^* = \mathbb{E}\left[\int_0^1 \lambda_i^{*-2}(t)\boldsymbol{\eta}_i(t)\boldsymbol{\eta}_i^{\mathsf{T}}(t)dN_t^i\right] = \mathbb{E}\left[\int_0^1 \lambda_i^{*-1}(t)\boldsymbol{\eta}_i(t)\boldsymbol{\eta}_i^{\mathsf{T}}(t)dt\right] = \mathbb{E}\left[\lambda_i^{*-1}\boldsymbol{\eta}_i\boldsymbol{\eta}_i^{\mathsf{T}}\right].$$

An example of the exponentially decaying kernel for the above quantities is given in Appendix **??**.

***Remark*** 1 (Multiple sequences). In practice, we may not be able to collect data long enough to achieve good confidence bound. Instead, we may have observations of many trials of independent Hawkes processes. We can write down the corresponding log-likelihood function and perform a similar analysis.

# 3 Main Results

We assume that the influence functions $\varphi_{ij}$ and background intensities $\mu_i$ are *given*, and our goal is to estimate the *unknown* parameters $\boldsymbol{\alpha}_i$. We estimate the unknown parameters $\boldsymbol{\alpha}_i$ by maximizing the log-likelihood function, i.e,

$$\widehat{\boldsymbol{\alpha}}_i := \underset{\boldsymbol{x} \in \mathbb{R}^D}{\arg\max}\, \ell_i(\boldsymbol{x}). \tag{6}$$

For each node $i$, by (1), $\lambda_i$ is linear in $\boldsymbol{\alpha}_i$, and by (2), we see that the log-likelihood function is concave in $\lambda_i$. Therefore, the MLE can be computed efficiently using convex optimization.

***Remark*** 2. In the general case, the Hessian matrix $H_i(\cdot)$ is negative definite everywhere when at least $D$ events happened on node $i$, and the MLE is unique. Indeed, when at least $D$ events happened in the time interval $[0, T)$, the vectors $\{\boldsymbol{\eta}_i(t) : dN_t^i = 1\}$ will have (in the general case) a linearly independent component of size $D$, and $H_i$ as the weighted sum of $\{-\boldsymbol{\eta}_i(t)\boldsymbol{\eta}_i^{\mathsf{T}}(t) : dN_t^i = 1, t < T\}$, is negative definite.

In this section, we present two different ways of uncertainty quantification for the MLE $\widehat{\boldsymbol{\alpha}}_i$. The first one is the classic asymptotic CI (for each entry $\alpha_{ij}$), which uses the fact that the MLE of such Hawkes processes is consistent and asymptotically normal. The second approach is our proposed method, which entails building a general confidence set for $\boldsymbol{\alpha}_i^*$ based on the more precise concentration-bound.

## 3.1 Asymptotic Confidence Intervals

It is known that the MLE of a temporal point process is asymptotically normal, and the empirical Fisher Information converges to the true Fisher Information with probability 1 [17]. Under our context, this translates into the following theorem.

**Theorem 3.1** (Asymptotic CI [17]). *Denote the empirical Fisher Information as*

$$\hat{I}_i(\widehat{\boldsymbol{\alpha}}_i) = -\frac{1}{T}\frac{\partial l_i(\widehat{\boldsymbol{\alpha}}_i)}{\partial \boldsymbol{\alpha}_i \partial \boldsymbol{\alpha}_i^{\mathsf{T}}} = \frac{1}{T}\int_0^T \hat{\lambda}_i^{-2}(t)\boldsymbol{\eta}_i(t)\boldsymbol{\eta}_i^{\mathsf{T}}(t)dN_t^i,$$

*then as $T \to \infty$,*

$$\sqrt{T}(\widehat{\boldsymbol{\alpha}}_i - \boldsymbol{\alpha}_i^*) \to \mathcal{N}(0, I_i^{*-1}),$$
$$\hat{I}_i(\widehat{\boldsymbol{\alpha}}_i) \to I_i^*.$$

Since our focus is on the CI for each $\alpha_{ij}$, Theorem 3.1 implies that as $T \to \infty$,

$$\sqrt{T}(\widehat{\alpha}_{ij} - \alpha_{ij}^*) \to \mathcal{N}(0, \sigma_{ij}^{*2}),$$

and

$$\widehat{\sigma}_{ij}^2(\widehat{\boldsymbol{\alpha}}_i) \to \sigma_{ij}^{*2},$$

where $\sigma_{ij}^{*2}$ is the $j$-th diagonal entry of $I_i^{*-1}$, $\widehat{\sigma}_{ij}^2(\widehat{\boldsymbol{\alpha}}_i)$ is the $j$-th diagonal entry of $\hat{I}_i^{-1}(\widehat{\boldsymbol{\alpha}}_i)$. An asymptotic CI on each entry $\alpha_{ij}$ is given by

$$\widehat{\alpha}_{ij} \pm Z_{\varepsilon/2D}\sqrt{\widehat{\sigma}_{ij}^2(\widehat{\boldsymbol{\alpha}}_i)/T},$$

where $Z_{\varepsilon/2D}$ is the corresponding percentage point for standard normal distribution, i,e., $\Phi(Z_{\varepsilon/2D}) = 1 - \varepsilon/2D$, and $\Phi(\cdot)$ is the cumulative distribution function of standard normal distribution.

## 3.2 Generalized confidence sets based on concentration bound

The classic asymptotic CI has a nice form and is easy to compute, but its confidence level has no guarantee when $T$ is not large enough. To be specific, there are three types of convergence involved in the asymptotic behavior of the MLE and the classic CI:

- the score function $S_i(\boldsymbol{\alpha}_i^*)$ is asymptotically normal,
- the empirical Hessian at $\boldsymbol{\alpha}_i^*$ converges to the Fisher Information,
- $\widehat{\boldsymbol{\alpha}}_i \to \boldsymbol{\alpha}_i^*$, and the statistical properties of $\widehat{\boldsymbol{\alpha}}_i$ converge to those of $\boldsymbol{\alpha}_i^*$.

In this section, we propose a general non-asymptotic confidence set for MLE $\widehat{\boldsymbol{\alpha}}_i$ by providing a concentration bound on the first type of convergence. In other words, we give the concentration bound on $S_i(\boldsymbol{\alpha}_i^*)$, which in turn provides the confidence set for $\boldsymbol{\alpha}_i^*$. This concentration result does not seem dependent on the convergence rate of the second term, and we further propose to use the third term's asymptotic behavior to approximate this confidence set and facilitate computation.

Our proof idea is as follows. We start with a similar step to proving the asymptotic result but follow with a tighter bound for the score function (the gradient of the log-likelihood function), leveraging a concentration bound for the continuous-time martingale.

First, we present the following general result. Since $\lambda_i^*(t)$ denotes the intensity function under true value $\boldsymbol{\alpha}_i^*$, we have the conditional expectation, given observations before time $t$, of $dN_t^i - \lambda_i^*(t)dt$ is 0, and

$$S_{i,t}(\boldsymbol{\alpha}_i^*) = \int_0^t \lambda_i^{*-1}(\tau)\boldsymbol{\eta}_i(\tau)(dN_\tau^i - \lambda_i^*(\tau)d\tau)$$

can be shown to be a *continuous-time martingale*.

The difficulty for a concentration bound here is that the variance of this process changes over time and cannot be bounded from above. Therefore, a standard Hoeffding or Bernstein type of concentration bound does not apply. Here we derive a concentration bound using the *intrinsic variance*, which depends on the data. Similar to [8], our intrinsic variance is a random process. Then we bound $S_i(\boldsymbol{\alpha}_i^*)$ in $K$ different directions and convert these concentration bounds into a confidence set for $\boldsymbol{\alpha}_i$.

**Theorem 3.2** (Confidence set for $\boldsymbol{\alpha}_i$). *Given data, for each $\boldsymbol{\alpha}_i$, let*

$$V_i(\boldsymbol{z}, \boldsymbol{\alpha}_i) = \int_0^T \left(\lambda_i(t)\exp(\lambda_i^{-1}(t)\boldsymbol{z}^{\mathsf{T}}\boldsymbol{\eta}_i(t)) - \boldsymbol{z}^{\mathsf{T}}\boldsymbol{\eta}_i(t) - \lambda_i(t)\right)dt. \qquad (7)$$

*For any given $\{\boldsymbol{z}_1, \ldots, \boldsymbol{z}_K\}$, a confidence set for $\boldsymbol{\alpha}_i$ at level $1 - \varepsilon$ is given by a polyhedron*

$$\mathcal{C}_{i,\varepsilon} = \left\{\boldsymbol{\alpha}_i \in \mathbb{R}^D : \forall k \in [K], \boldsymbol{z}_k^{\mathsf{T}}S_i(\boldsymbol{\alpha}_i) - V_i(\boldsymbol{z}_k, \boldsymbol{\alpha}_i) \leq \ln(K/\varepsilon)\right\}. \qquad (8)$$

**UQ for each** $\alpha_{ij}$. Based on the confidence set for vector $\boldsymbol{\alpha}_i$, we can construct the CI for each entry. So, naturally, we would like our confidence set $\mathcal{C}_{i,\varepsilon}$ to resemble an orthotope parallel to the axes. Based on the mean value theorem, for any $\boldsymbol{\alpha}_i$ in the confidence set, there exists $\tilde{\boldsymbol{\alpha}}_i$ between $\boldsymbol{\alpha}_i$ and $\widehat{\boldsymbol{\alpha}}_i$ such that

$$S_i(\boldsymbol{\alpha}_i) - S_i(\hat{\boldsymbol{\alpha}}_i) = H_i(\tilde{\boldsymbol{\alpha}}_i)(\boldsymbol{\alpha}_i - \widehat{\boldsymbol{\alpha}}_i).$$

Since $\widehat{\boldsymbol{\alpha}}_i$ is the maximum likelihood estimate, we have $S_i(\widehat{\boldsymbol{\alpha}}_i) = 0$. The confidence set is supposed to be a small neighborhood of $\boldsymbol{\alpha}_i^*$, and when $T$ is large, we have

$$S_i(\boldsymbol{\alpha}_i) = H_i(\tilde{\boldsymbol{\alpha}}_i)(\boldsymbol{\alpha}_i - \widehat{\boldsymbol{\alpha}}_i) \approx T I_i^*(\boldsymbol{\alpha}_i - \widehat{\boldsymbol{\alpha}}_i).$$

Intuitively, we can let $K = 2D$, $\boldsymbol{z}_1, \cdots, \boldsymbol{z}_{2D}$ be in the same direction with the columns of $\pm I_i^{*-1}$, each is supposed to correspond to the upper/lower bound on the entries of $\boldsymbol{\alpha}_i$, and the confidence set $\mathcal{C}_{i,\varepsilon}$ will approximately be a box around the MLE. Formally speaking, we have the following lemma.

**Lemma 3.1.** *Under the assumption that the moment generating function of $\boldsymbol{\eta}_i$ exists, there exists a neighborhood $U$ of $\mathbf{0}$, such that*

$$\|S_i(\boldsymbol{\alpha}_i) - T I_i^*(\boldsymbol{\alpha}_i - \widehat{\boldsymbol{\alpha}}_i)\| \le O(T)\|\boldsymbol{\alpha}_i - \widehat{\boldsymbol{\alpha}}_i\|^2 + o(T)\|\boldsymbol{\alpha}_i - \widehat{\boldsymbol{\alpha}}_i\|, \tag{9}$$

*and*

$$\left| V_i(\boldsymbol{z}, \boldsymbol{\alpha}_i) - \frac{T}{2}\boldsymbol{z}^\mathsf{T} I_i^* \boldsymbol{z} \right| \le o(T)\|\boldsymbol{z}\|^2 + O(T)\|\boldsymbol{\alpha}_i - \widehat{\boldsymbol{\alpha}}_i\|\|\boldsymbol{z}\|^2 + O(T)\|\boldsymbol{z}\|^3, \tag{10}$$

*uniformly for any $\boldsymbol{\alpha}_i \ge 0$, $\boldsymbol{z} \in U$ with high probability. Above, $\|\cdot\|$ denotes $\ell_2$ norm.*

To make the confidence set at level $1 - \varepsilon$ as small as possible, we can choose $\boldsymbol{z}$ as the following:

**Proposition 1.** *Let $K = 2D$, and $\boldsymbol{z}_1, \cdots, \boldsymbol{z}_{2D}$ be*

$$\pm \sqrt{\frac{2\ln(2D/\varepsilon)}{T\sigma_{ij}^{*2}}} I_i^{*-1} \boldsymbol{e}_j, \ j = 1, \cdots, D, \tag{11}$$

*respectively. As $T \to \infty$, for any $j \in [D]$, the width of $\mathcal{C}_{i,\varepsilon}$ in the direction of $\alpha_{ij}$ is $2\sqrt{2\ln(2D/\varepsilon)\sigma_{ij}^{*2}/T}(1 + o(1))$ with probability 1.*

Note that the width of $\mathcal{C}_{i,\varepsilon}$ is asymptotically a constant times the classic asymptotic confidence intervals.

### 3.3 Concentration confidence bound with adapted $z$

In reality, we don't have the true parameter $\boldsymbol{\alpha}_i^*$ or the Fisher Information $I_i^*$. So to make the confidence set as small as possible, we will have to estimate the Fisher Information. The challenge is that we cannot estimate the Fisher Information by simulation, because we don't know the true parameter, and simulation using the MLE will make our choice of $\boldsymbol{z}_k$ depend on the data. What we can do, though, is use data that comes earlier to estimate the Fisher Information $I_i^*$ and the proper choice of $\boldsymbol{z}$ for future data. This leads to our concentration bound with adapted $\boldsymbol{z}$:

**Theorem 3.3** (Martingale concentration for score function with adapted $\boldsymbol{z}$). *For any measurable process $(\boldsymbol{z}(t) \in \mathbb{R}^D)_{t=0}^T$ adapted to $(\mathcal{H}_{t^-})_{t=0}^T$, where $(\mathcal{H}_t)_{t=0}^T$ is the filtration of the Hawkes process, any $\varepsilon \in (0, 1)$, we have*

$$\Pr\left( \int_0^T \boldsymbol{z}^\mathsf{T}(t) dS_{i,t}(\boldsymbol{\alpha}_i^*) - V_i(\boldsymbol{z}, \boldsymbol{\alpha}_i^*) \ge \ln(1/\varepsilon) \right) \le \varepsilon, \tag{12}$$

*where*

$$dS_{i,t}(\boldsymbol{\alpha}_i) = \lambda_i(t)^{-1}\boldsymbol{\eta}_i(t)(dN_t^i - \lambda_i(t)dt),$$

*and*

$$V_i(\boldsymbol{z}, \boldsymbol{\alpha}_i) = \int_0^T \log\left( \mathbb{E}\left( \exp\left(\boldsymbol{z}^\mathsf{T}(t) dS_{i,t}(\boldsymbol{\alpha}_i)\right) \big| \mathcal{H}_{t^-} \right) \right)$$

$$= \int_0^T \left( \lambda_i(t) \exp\left( \lambda_i^{-1}(t)\boldsymbol{z}^\mathsf{T}(t)\boldsymbol{\eta}_i(t) \right) - \boldsymbol{z}^\mathsf{T}(t)\boldsymbol{\eta}_i(t) - \lambda_i(t) \right) dt.$$

**Corollary 1** (UQ for each $\alpha_{ij}$). *For any $\alpha_i$, $t \in [0, T]$, let $\hat{I}_i(\alpha_i, t)$ be some estimator for the Fisher Information given data up to time $t^-$. Let $z_1(t, \alpha_i), \cdots, z_{2D}(t, \alpha_i)$ be*

$$\pm \sqrt{\frac{2 \ln(2D/\varepsilon)}{T e_j^\intercal \hat{I}_i^{-1}(\alpha_i, t) e_j}} \hat{I}_i^{-1}(\alpha_i, t) e_j, \ j = 1, \cdots, D,$$

*respectively. Then*

$$\mathcal{C}_{i,\varepsilon} = \left\{ \alpha_i \in \mathbb{R}^D : \int_0^T z_k^\intercal(t, \alpha_i) dS_{i,t}(\alpha_i) - V_i(z_k, \alpha_i) \leq \ln(2D/\varepsilon), k = 1, \cdots, 2D \right\}$$

*is a confidence set for $\alpha_i$ at level $1 - \varepsilon$.*

***Remark*** 3. An example of estimator for the Fisher Information is

$$\hat{I}_i(\alpha_i, t) = -\frac{1}{t} \int_0^t \lambda_i^{-2}(\tau) \eta_i(\tau) \eta_i^\intercal(\tau) dN_\tau^i,$$

and if the estimator is rank deficient, we simply take it to be the identity matrix.

For simplicity, we use $g_k(\alpha_i)$ to denote

$$\int_0^T z_k^\intercal(t, \alpha_i) dS_{i,t}(\alpha_i) - V_i(z_k, \alpha_i), \ k = 1, \cdots, 2D.$$

The CI of entry $\alpha_{ij}$ is then $\{\alpha_{ij} : g_k(\alpha_i) \leq \ln(2D/\varepsilon), k = 1, \cdots, 2D\}$, This CI can be computed by numerically inverting the functions $g_k, k = 1, \cdots, 2D$, but it may be time-consuming. Since $\widehat{\alpha}_i \to \alpha_i^*$ with probability one when $T \to \infty$, we can approximate $g_k(\alpha_i^*)$ using first order Taylor expansion at $\widehat{\alpha}_i$. Let

$$\tilde{g}_k(\alpha_i) = g_k(\widehat{\alpha}_i) + (\alpha_i - \widehat{\alpha}_i)^\intercal \frac{\partial g_k(\widehat{\alpha}_i)}{\partial \alpha_i},$$

an approximated confidence set for $\alpha_i$ is

$$\mathcal{C}_{i,\varepsilon}^p = \left\{ \alpha_i \in \mathbb{R}^D : \tilde{g}_k(\alpha_i) \leq \ln(2D/\varepsilon), k = 1, \cdots, 2D \right\},$$

which is a polyhedron.

With the polyhedron $\mathcal{C}_{i,\varepsilon}^p$, we can easily get CI on each entry $\alpha_{ij}$

$$\left[ \min\{\alpha_{ij} : \alpha_i \in \mathcal{C}_{i,\varepsilon}^p\}, \max\{\alpha_{ij} : \alpha_i \in \mathcal{C}_{i,\varepsilon}^p\} \right].$$

using linear optimization.

Algorithm 1 summarizes how to find the concentration-bound based confidence set.

---

**Algorithm 1:** Polyhedral Confidence Set for $\alpha_i$

---

**Input: confidence level** $1 - \varepsilon$**, data** $\{(t_i, u_i)\}$**, estimator** $\hat{I}_i^{-1}(\cdot, \cdot)$**;**
**Compute the MLE** $\widehat{\alpha}_i$ **by convex optimization** (6)**;**
**for** $k = 1, \cdots, 2D$ **do**

$$g_k(\widehat{\alpha}_i) = \int_0^T dS_{i,t}(\widehat{\alpha}_i) z_k(t, \widehat{\alpha}_i) - V_i(z_k, \widehat{\alpha}_i),$$

$$g_k'(\widehat{\alpha}_i) := \frac{\partial g_k(\widehat{\alpha}_i)}{\partial \alpha_i}.$$

**end**
**Output:** $\mathcal{C}_{i,\varepsilon}^p := \left\{ \alpha_i \in \mathbb{R}^D : g_k(\widehat{\alpha}_i) + (\alpha_i - \widehat{\alpha}_i)^\intercal g_k'(\widehat{\alpha}_i) \leq \ln(2D/\varepsilon), k = 1, \cdots, 2D \right\}$

---

## 4 Numerical Experiment

In this section, we present a numerical example based on synthetic data to demonstrate the performance of the proposed confidence intervals. We compare the coverage ratio of the confidence intervals: the percentage of confidence intervals that contain the true parameters, for the same nominal confidence level $(1 - \varepsilon)$.

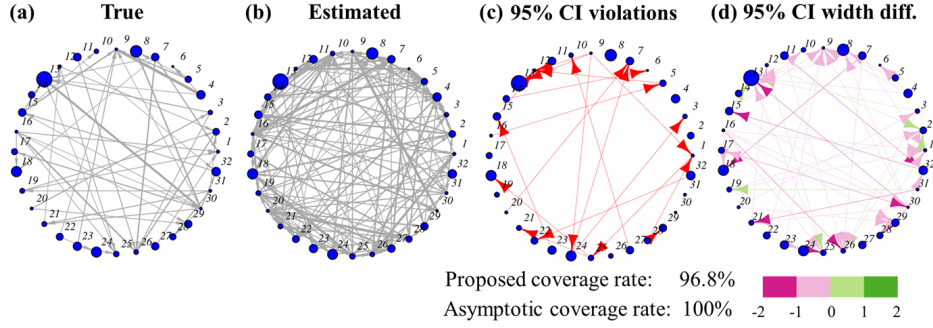

**(a)** True     **(b)** Estimated     **(c) 95% CI violations**     **(d) 95% CI width diff.**

Proposed coverage rate: 96.8%
Asymptotic coverage rate: 100%

Figure 1: (a) and (b): Visualizing the "true" and estimated influence matrix $A$ as edges and background rates $\mu$ as nodes. Wider edges indicate greater influence, and larger nodes indicate greater background rates. (c): Edges whose 95% CIs do not cover the true influence parameter for the proposed CI method. The coverage rate for the proposed and asymptotic CIs are 96.8% and 100%, respectively. (d): Visualizing the difference in 95% CI widths between the proposed and asymptotic CIs. Purple edges and green edges indicate narrower and wider widths for the proposed CI, respectively.

We study uncertainty quantification for reconstructing neuronal networks. Recent developments in neural engineering have allowed researchers to simultaneously record precise spike train data from large numbers of biological neurons [12]. A key challenge is harnessing this data to learn the connectivity of biological neural networks, which provides insight on the functions of such networks. We show next how the proposed method can quantify the uncertainty of the reconstructed neuronal connectivity from spiking data. This uncertainty is crucial for neuronal reconstruction: it provides a principled statistical framework for testing different neurological theories and hypotheses.

The experimental set-up is as follows. The neural spike train data is simulated via the PyNN Python package [5] with the NEURON simulator [4], which was chosen over in vivo recordings for straightforward data collection. The neuronal network consists of excitatory and inhibitory networks in a ratio of 4 to 1, which are connected sparsely and at random. The neurons are modeled as exponential integrate-and-fire neurons with default parameters, which have been shown to accurately capture biological neural dynamics [2]. Following [3], each excitatory neuron receives a stochastic Poisson process-distributed excitation from an external source, reflecting the external inputs from biological networks either from the environment or from neurons which are not being recorded.

Using the above network structure with $D = 32$ neurons, we simulate a *long* sequence (2000 seconds) of spiking data, and fit a Hawkes network using an exponential influence function with a decay rate of 1 millisecond. This fitted model (with estimates of the influence matrix $A$ and background rate vector $\mu$) can be viewed as the Hawkes network "closest" to the complex neuroscience model which generated the data. The fitted parameters for $A$ and $\mu$ (see Figure 1 (a)) are then set as the "true" parameters for evaluating CI coverage. We then simulate a *shorter* sequence (400 seconds) of spiking data for constructing the proposed (non-asymptotic) and asymptotic CIs on $A$. Figure 1 (b) shows the MLE of $A$, estimated using this shorter sequence. Note that, while the connectivity for the "true" topology is quite sparse, the estimated connectivity is noticeably more dense, perhaps due to the limited data in the shorter sequence. In this limited data setting, there is an increasing need for uncertainty quantification to validate neuronal connectivity.

Consider now the coverage performance of the proposed (non-asymptotic) and asymptotic CIs. At a confidence level of 95%, the coverage rate of the proposed method (over all influence parameters in $A$) is 96.8%, whereas the coverage rate for the asymptotic method is 100%. Hence, the proposed CIs indeed provide similar coverage to the desired confidence level of 95%, whereas the asymptotic CIs are too wide and over-covers the true parameters. Figure 1 shows the edges with influence parameters *not* covered by the proposed method. All of these edges have a true influence of 0, i.e., such edges were not in the true topology, but had positive CIs. Figure 1 visualizes the difference in CI widths between the proposed and asymptotic CIs, for edges with non-zero true influence. Here, purple edges and green edges indicate narrower and wider widths for the proposed CI, respectively. We see that the proposed method yields noticeably narrower CIs compared to the asymptotic approach, which enables more precise inference on the influence matrix. This in turn provides greater certainty on the reconstructed neuronal network, particularly given limited experimental data.

## Broader Impact

Our method can be useful for many applications involving Hawkes processes, including seismology, social networks, neuroscience and more. In particular, it is useful for performing causal inference and making statistically significant claims. Recent developments in neuroscience and engineering have allowed researchers to simultaneously record precise spiking data from large numbers of biological neurons. A key challenge is harnessing this experimental data to learn the underlying connectivity of biological neural networks, which is integral for understanding the functions of such networks. We show how the proposed model can be used to both learn this connectivity information and quantify uncertainty from observed neural spike data.

## Acknowledgement

This work is partially funded by an NSF CAREER Award CCF-1650913, CMMI-2015787, DMS-1938106, and DMS-1830210.

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
