[Supplementary Material]

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

\boldsymbol{e}_j^{\mathsf{T}}\hat{I}_i^{-1}(\boldsymbol{\alpha}_i,t)\boldsymbol{e}_j}}\hat{I}_i^{-1}(\boldsymbol{\alpha}_i,t)\boldsymbol{e}_j, \; j = 1, \cdots, D,$$

*respectively. Then*

$$\mathcal{C}_{i,\varepsilon} = \left\{ \boldsymbol{\alpha}_i \in \mathbb{R}^D : \int_0^T \boldsymbol{z}_k^{\mathsf{T}}(t, \boldsymbol{\alpha}_i)dS_{i,t}(\boldsymbol{\alpha}_i) - V_i(\boldsymbol{z}_k, \boldsymbol{\alpha}_i) \leq \ln(2D/\varepsilon), k = 1, \cdots, 2D \right\}$$

*is a confidence set for $\boldsymbol{\alpha}_i$ at level $1 - \varepsilon$.*

***Remark** 3.* An example of estimator for the Fisher Information is

$$\hat{I}_i(\boldsymbol{\alpha}_i, t) = -\frac{1}{t}\int_0^t \lambda_i^{-2}(\tau)\boldsymbol{\eta}_i(\tau)\boldsymbol{\eta}_i^{\mathsf{T}}(\tau)dN_\tau^i,$$

and if the estimator is rank deficient, we simply take it to be the identity matrix.

For simplicity, we use $g_k(\boldsymbol{\alpha}_i)$ to denote

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

# A    Example: Exponential decay function

Here we give the analysis for the score function and the Fisher information under exponential decay function $\varphi_{ij}(\Delta t) = \beta e^{-\beta \Delta t}$. The score function is,

$$S_i(\boldsymbol{\alpha}_i^*) = \int_0^T \frac{\int_0^t \beta e^{-\beta(t-\tau)} d\boldsymbol{N}_\tau}{\mu_i + (\boldsymbol{\alpha}_i^*)^\intercal \int_0^t \beta e^{-\beta(t-\tau)} d\boldsymbol{N}_\tau} (dN_t^i - \lambda_i^*(t)dt),$$

where $d\boldsymbol{N}_t = (dN_t^1, \cdots dN_t^D)^\intercal$.

We show that $S_i(\boldsymbol{\alpha}_i^*)$ is small by giving an upper bound of its covariance matrix, which is $T$ times the Fisher information.

Assume the Hawkes process with parameter $\boldsymbol{\alpha}_i, \mu_i, i = 1, \cdots, D, \beta$ is stationary, we have

$$I_i^* = \mathbb{E} \left[ \frac{\left( \int_{-\infty}^t \beta e^{-\beta(t-\tau)} d\boldsymbol{N}_\tau \right) \left( \int_{-\infty}^t \beta e^{-\beta(t-\tau)} d\boldsymbol{N}_\tau \right)^\intercal}{\mu_i + (\boldsymbol{\alpha}_i^*)^\intercal \int_{-\infty}^t \beta e^{-\beta(t-\tau)} d\boldsymbol{N}_\tau} \right]$$

Since $\boldsymbol{\alpha}_i^T \int_{-\infty}^t \beta e^{-\beta(t-\tau)} dN_\tau \geq 0$, we have

$$I_i^* \preceq \mu_i^{-1} \underbrace{\mathbb{E} \left[ \left( \int_{-\infty}^t \beta e^{-\beta(t-\tau)} d\boldsymbol{N}_\tau \right) \left( \int_{-\infty}^t \beta e^{-\beta(t-\tau)} d\boldsymbol{N}_\tau \right)^\intercal \right]}_{W},$$

where $W$ has a close-form expression for Hawkes processes with exponential influence function, derived from [2] and [16].

**Lemma A.1.**

$$W = \Lambda\Lambda^\intercal + \frac{\beta}{2}\Sigma + \frac{\beta}{4}A(\mathbb{I} - A)^{-1}\Sigma + \frac{\beta}{4}\Sigma A^\intercal (\mathbb{I} - A^\intercal)^{-1},$$

*where $\mathbb{I}$ is the identity matrix, $A = (\boldsymbol{\alpha}_1^*, \cdots, \boldsymbol{\alpha}_D^*)^\intercal$, $\Lambda = (\mathbb{I} - A)^{-1}\boldsymbol{\mu}$ is the expected intensity, and $\Sigma = diag(\Lambda)$.*

*Proof.* By Lemma 2 and 3 in [16], we have

$$\mathbb{E}[d\boldsymbol{N}_t] = \Lambda dt,$$

and

$$\text{Cov}[d\boldsymbol{N}_t, d\boldsymbol{N}_{t'}^\intercal] = c(t - t')dtdt',$$

where

$$c(\tau) = \begin{cases} \beta e^{-\beta(\mathbb{I}-A)\tau} A \left( \mathbb{I} + \frac{1}{2}(\mathbb{I} - A)^{-1}A \right) \Sigma, & \tau > 0; \\ \Sigma\delta(\tau), & \tau = 0; \\ c(-\tau)^\intercal, & \tau < 0, \end{cases}$$

where $\delta(\cdot)$ is the Dirac delta function. Then

$$W = \mathbb{E} \left[ \left( \int_{-\infty}^t \beta e^{-\beta(t-\tau)} d\boldsymbol{N}_\tau \right) \left( \int_{-\infty}^t \beta e^{-\beta(t-\tau)} d\boldsymbol{N}_\tau \right)^\intercal \right]$$

$$= \mathbb{E} \left[ \int_{-\infty}^0 \int_{-\infty}^0 \beta^2 e^{\beta(t+t')} d\boldsymbol{N}_t d\boldsymbol{N}_{t'}^\intercal \right]$$

$$= \int_{-\infty}^0 \int_{-\infty}^0 \beta^2 e^{\beta(t+t')} \mathbb{E}[d\boldsymbol{N}_t d\boldsymbol{N}_{t'}]$$

$$= \Lambda\Lambda^\intercal + \int_{-\infty}^0 \int_{-\infty}^0 \beta^2 e^{\beta(t+t')} \text{Cov}[d\boldsymbol{N}_t, d\boldsymbol{N}_{t'}]$$

$$= \Lambda\Lambda^\intercal + \int_{-\infty}^0 \beta^2 e^{2\beta t}\Sigma dt + \iint_{t \leq 0, \tau \in (0, -t]} \beta^2 e^{\beta(2t+\tau)}(c(\tau) + c(-\tau))dtd\tau$$

$$
= \Lambda\Lambda^\mathsf{T} + \frac{\beta}{2}\Sigma + \int_0^\infty (c(\tau) + c(-\tau))d\tau \int_{-\infty}^{-\tau} \beta^2 e^{\beta(2t+\tau)} dt
$$

$$
= \Lambda\Lambda^\mathsf{T} + \frac{\beta}{2}\Sigma + \int_0^\infty (c(\tau) + c(-\tau))\frac{\beta}{2}e^{-\beta\tau} d\tau
$$

$$
= \Lambda\Lambda^\mathsf{T} + \frac{\beta}{2}\Sigma + \int_0^\infty \frac{\beta^2}{2}\left[ e^{-\beta(2\mathbb{I}-A)\tau} A(\mathbb{I} + \frac{1}{2}(\mathbb{I}-A)^{-1}A)\Sigma \right.
$$

$$
\left. + \left( e^{-\beta(2\mathbb{I}-A)\tau} A(\mathbb{I} + \frac{1}{2}(\mathbb{I}-A)^{-1}A)\Sigma \right)^\mathsf{T} \right] d\tau
$$

$$
= \Lambda\Lambda^\mathsf{T} + \frac{\beta}{2}\Sigma + \frac{\beta}{2}\left[ (2\mathbb{I}-A)^{-1}A(\mathbb{I} + \frac{1}{2}(\mathbb{I}-A)^{-1}A)\Sigma \right.
$$

$$
\left. + \left( (2\mathbb{I}-A)^{-1}A(\mathbb{I} + \frac{1}{2}(\mathbb{I}-A)^{-1}A)\Sigma \right)^\mathsf{T} \right].
$$

We notice that

$$
(2\mathbb{I}-A)^{-1}A(\mathbb{I} + \frac{1}{2}(\mathbb{I}-A)^{-1}A) = (2\mathbb{I}-A)^{-1}A(\mathbb{I}-A)^{-1}(\mathbb{I}-A/2) = \frac{1}{2}A(\mathbb{I}-A)^{-1}.
$$

Together, we prove the lemma. $\qquad\square$

**Lemma A.2.** *For any vector $z$,*

$$
P\left( z^\mathsf{T} S_i(\alpha_i^*) \geq \varepsilon\sqrt{T} \right) \leq \frac{\mu_i^{-1} z^\mathsf{T} W z}{\varepsilon^2}.
$$

*Proof.*

$$
\mathrm{Var}\left[ z^\mathsf{T} S_i(\alpha_i^*) \right] = T z^\mathsf{T} I_i^* z \leq \mu_i^{-1} T z^\mathsf{T} W z,
$$

by Markov's inequality on the random variable $(z^\mathsf{T} S_i(\alpha_i^*))^2$, we proof the lemma.

$\qquad\square$

# B  Proofs

The proof of Theorem 3.2 and Theorem 3.3 is an immediate results of the following two lemmas.

**Lemma B.1.** *For any measurable random process $(z(t) \in \mathbb{R}^D)_{t\in[0,T]}$ adapted to the same filtration $(\mathcal{H}_t)_{t\in[0,T]}$ with the Hawkes process, let the intrinsic variance of $\int_0^t z^\mathsf{T}(\tau)dS_{i,\tau}(\alpha_i^*)$ (denoted by $V_{i,t}(z)$) be a random process also adapted to $(\mathcal{H}_t)_{t\in[0,T]}$, such that there exists a supermartingale $(M_t(z))_{t\in[0,T]}$ with respect to $(\mathcal{H}_t)_{t\in[0,T]}$,*

$$
\exp\left( \int_0^t z^\mathsf{T}(\tau)dS_{i,\tau}(\alpha_i^*) - V_{i,t}(z) \right) \leq M_t(z)
$$

*almost surely. Then $\forall \varepsilon \in (0,1)$,*

$$
\mathrm{Pr}\left( \int_0^T z^\mathsf{T}(t)dS_{i,t}(\alpha_i^*) - V_{i,T}(z) \geq \ln(\mathbb{E}[M_0(z)]/\varepsilon) \right) \leq \varepsilon.
$$

*Proof.* By the property of a supermartingale, we have

$$
\mathbb{E}\left[ \exp\left( \int_0^T z^\mathsf{T}(t)dS_{i,t}(\alpha_i^*) - V_{i,T}(z) \right) \right] \leq \mathbb{E}[M_T(z)] \leq \mathbb{E}[M_0(z)],
$$

and by Markov's inequality,

$$\Pr\left[\int_0^T \boldsymbol{z}^\mathsf{T}(t)dS_{i,t}(\boldsymbol{\alpha}_i^*) - V_{i,T}(\boldsymbol{z}) \geq \ln(\mathbb{E}[M_0(\boldsymbol{z})]/\varepsilon)\right]$$

$$= \Pr\left[\exp\left(\int_0^T \boldsymbol{z}^\mathsf{T}(t)dS_{i,t}(\boldsymbol{\alpha}_i^*) - V_{i,T}(\boldsymbol{z})\right) \geq \mathbb{E}[M_0(\boldsymbol{z})]/\varepsilon\right]$$

$$\leq \frac{\mathbb{E}\left[\exp\left(\int_0^T \boldsymbol{z}^\mathsf{T}(t)dS_{i,t}(\boldsymbol{\alpha}_i^*) - V_{i,T}(\boldsymbol{z})\right)\right]}{\mathbb{E}[M_0(\boldsymbol{z})]/\varepsilon} \leq \varepsilon.$$

$\square$

Moreover, the intrinsic variance can be characterized explicitly by the following result.

**Lemma B.2.** *Let*

$$V_{i,t}(\boldsymbol{z}) = \int_0^t \left(\lambda_i^*(\tau)\exp(\lambda_i^{*-1}(\tau)\boldsymbol{z}^\mathsf{T}(\tau)\boldsymbol{\eta}_i(\tau)) - \boldsymbol{z}^\mathsf{T}(\tau)\boldsymbol{\eta}_i(\tau) - \lambda_i^*(\tau)\right)d\tau. \qquad (13)$$

$$M_t(\boldsymbol{z}) = \exp\left(\int_0^t \boldsymbol{z}^\mathsf{T}(\tau)dS_{i,\tau}(\boldsymbol{\alpha}_i^*) - V_{i,t}(\boldsymbol{z})\right)$$

*is a supermartingale, with $M_0(\boldsymbol{z}) = 1$ almost surely.*

*Proof.* For any $t$, since $V_{i,t}$ is continuous and the right derivative exists,

$$\lim_{\Delta t \to 0^+} \frac{\log \mathbb{E}\left[M_{t+\Delta t}(\boldsymbol{z})/M_t(\boldsymbol{z})|\mathcal{H}_t\right]}{\Delta t}$$

$$= \lim_{\Delta t \to 0^+} \frac{\log \mathbb{E}\left[\exp\left(\int_t^{t+\Delta t} \boldsymbol{z}^\mathsf{T}(\tau)dS_{i,\tau}(\boldsymbol{\alpha}_i^*) - \Delta V_{i,t}(\boldsymbol{z})\right)|\mathcal{H}_t\right]}{\Delta t}$$

$$= \lim_{\Delta t \to 0^+} \frac{\log \mathbb{E}\left[\exp\left(\boldsymbol{z}^\mathsf{T}(t)\boldsymbol{\eta}_i(t)(\lambda_i^{*-1}\Delta N_t^i - \Delta t)|\mathcal{H}_t\right]\right.}{\Delta t} - (\lambda_i^*\exp(\lambda_i^{*-1}\boldsymbol{z}^\mathsf{T}(t)\boldsymbol{\eta}_i(t)) - \boldsymbol{z}^\mathsf{T}(t)\boldsymbol{\eta}_i(t) - \lambda_i^*(t))$$

$$= \lim_{\Delta t \to 0^+} \frac{\log \left(\lambda_i^*(t)\Delta t \exp(\lambda_i^{*-1}\boldsymbol{z}^\mathsf{T}(t)\boldsymbol{\eta}_i(t)) + (1 - \lambda_i^*(t)\Delta t)\exp(-\boldsymbol{z}^\mathsf{T}(t)\boldsymbol{\eta}_i(t)\Delta t)\right)}{\Delta t}$$

$$- (\lambda_i^*\exp(\lambda_i^{*-1}\boldsymbol{z}^\mathsf{T}(t)\boldsymbol{\eta}_i(t)) - \boldsymbol{z}^\mathsf{T}(t)\boldsymbol{\eta}_i(t) - \lambda_i^*(t))$$

$$= 0.$$

From this we can see that $M_t$ is actually a martingale. $\square$

*Proof of Theorem 3.2, Theorem 3.3 and Corollary 1.* From Lemma B.1 and Lemma B.2, we have immediately

$$\Pr\left[\int_0^T \boldsymbol{z}^\mathsf{T}(t)dS_{i,t}(\boldsymbol{\alpha}_i^*) - V_{i,T}(\boldsymbol{z}) \geq \ln(1/\varepsilon)\right] \leq \varepsilon, \ \forall \boldsymbol{z} \in \mathbb{R}^D, \forall \varepsilon \in (0,1), \qquad (14)$$

where $V_{i,T}(\boldsymbol{z})$ is chosen as (13). Moreover, we can also choose multiple $\boldsymbol{z}$ to bound $S_i(\boldsymbol{\alpha}_i^*)$ in all directions. By simple union bound, it holds that

$$\Pr\left[\exists k \in [K], \int_0^T \boldsymbol{z}_k^\mathsf{T}(t)dS_{i,t}(\boldsymbol{\alpha}_i^*) - V_{i,T}(\boldsymbol{z}_k) \geq \ln(K/\varepsilon)\right] \leq K\varepsilon/K = \varepsilon, \ \forall \boldsymbol{z}_1, \cdots, \boldsymbol{z}_K \in \mathbb{R}^D, \forall \varepsilon \in (0,1).$$

We define continuous process $V_{i,t}$ for any $\boldsymbol{\alpha}_i$ as (7), $\boldsymbol{\alpha}_i^*$ falls into the confidence set $\mathcal{C}_{i,\varepsilon}$ with probability at least $1 - \varepsilon$. $\square$

The proof of Lemma 3.1 relies on the following lemma:

**Lemma B.3** (Ogata [18], Lemma 2). *If $\xi_t$ is a stationary predictable process, then*

$$\frac{1}{T}\int_0^T \xi_t dt \to \mathbb{E}\left[\xi\right]$$

*with probability 1. In addition, if $\xi_t$ has finite second moment, then*

$$\frac{1}{T}\int_0^T \xi_t \frac{dN_t}{\lambda(t)} \to \mathbb{E}\left[\xi\right]$$

*with probability 1.*

*Proof of Lemma 3.1.* Denote

$$\Delta\boldsymbol{\alpha}_i = \boldsymbol{\alpha}_i - \hat{\boldsymbol{\alpha}}_i.$$

Using Taylor expansion and based on the mean value theorem, there exists $\tilde{\boldsymbol{\alpha}}_i$ between $\boldsymbol{\alpha}_i$ and $\hat{\boldsymbol{\alpha}}_i$, such that

$$\frac{1}{T}S_i(\boldsymbol{\alpha}_i) = \frac{1}{T}H_i(\hat{\boldsymbol{\alpha}}_i)\Delta\boldsymbol{\alpha}_i + \frac{1}{2T}\sum_{k,l\in[D]}\frac{\partial^2 S_i(\tilde{\boldsymbol{\alpha}}_i)}{\partial\alpha_{ik}\partial\alpha_{il}}\Delta\alpha_{ik}\Delta\alpha_{il}.$$

For any $j,k,l \in [D]$, the $j$-th entry of $\frac{\partial^2 S_i(\tilde{\boldsymbol{\alpha}}_i)}{\partial\alpha_{ik}\partial\alpha_{il}}$ is

$$\int_0^T \frac{2\eta_{ij}(t)\eta_{ik}(t)\eta_{il}(t)}{\tilde{\lambda}_i^3(t)}dN_t^i.$$

Under the assumption that the moment generating function of $\boldsymbol{\eta}_i$ exists, and $\tilde{\lambda}_i \geq \mu_i > 0$, by Lemma B.3, as $T \to \infty$,

$$\frac{1}{T}\int_0^T \frac{2\eta_{ij}(t)\eta_{ik}(t)\eta_{il}(t)}{\tilde{\lambda}_i^3(t)}dN_t^i \leq \frac{1}{T\mu_i^3}\int_0^T 2\eta_{ij}(t)\eta_{ik}(t)\eta_{il}(t)dN_t^i$$

is uniformly bounded for any $\tilde{\boldsymbol{\alpha}}_i \geq 0$ with probability 1. Now we have for any $\boldsymbol{\alpha}_i$,

$$\|S_i(\boldsymbol{\alpha}_i) - H_i(\hat{\boldsymbol{\alpha}}_i)\Delta\boldsymbol{\alpha}_i\| \leq O(T)\|\Delta\boldsymbol{\alpha}_i\|^2.$$

Since

$$\frac{1}{T}H_i(\hat{\boldsymbol{\alpha}}_i) \to I_i^*$$

with probability 1, we have

$$\|S_i(\boldsymbol{\alpha}_i) - TI_i^*\Delta\boldsymbol{\alpha}_i\| \leq O(T)\|\Delta\boldsymbol{\alpha}_i\|^2 + \|(H_i(\hat{\boldsymbol{\alpha}}_i) - TI_i^*)\Delta\boldsymbol{\alpha}_i\| \leq O(T)\|\Delta\boldsymbol{\alpha}_i\|^2 + o(T)\|\Delta\boldsymbol{\alpha}_i\|,$$

which is (9).

For (10), similarly we use the Taylor expansion at $\boldsymbol{z} = 0$ and $\hat{\boldsymbol{\alpha}}_i$, by the mean value theorem,

$$V_i(\boldsymbol{z}, \boldsymbol{\alpha}_i) = V_i(\boldsymbol{0}, \boldsymbol{\alpha}_i) + \frac{\partial V_i(\boldsymbol{0}, \boldsymbol{\alpha}_i)}{\partial\boldsymbol{z}^\mathsf{T}}\boldsymbol{z} + \frac{1}{2}\boldsymbol{z}^\mathsf{T}\frac{\partial^2 V_i(\boldsymbol{0}, \boldsymbol{\alpha}_i)}{\partial\boldsymbol{z}\partial\boldsymbol{z}^\mathsf{T}}\boldsymbol{z} + \frac{1}{6}\sum_{j,k,l\in[D]}\frac{\partial^3 V_i(\tilde{\boldsymbol{z}}, \boldsymbol{\alpha}_i)}{\partial z_j\partial z_k\partial z_l}z_j z_k z_l$$

$$= \frac{1}{2}\boldsymbol{z}^\mathsf{T}\frac{\partial^2 V_i(\boldsymbol{0}, \boldsymbol{\alpha}_i)}{\partial\boldsymbol{z}\partial\boldsymbol{z}^\mathsf{T}}\boldsymbol{z} + \frac{1}{6}\sum_{j,k,l\in[D]}\frac{\partial^3 V_i(\tilde{\boldsymbol{z}}, \boldsymbol{\alpha}_i)}{\partial z_j\partial z_k\partial z_l}z_j z_k z_l$$

$$= \frac{1}{2}\boldsymbol{z}^\mathsf{T}\frac{\partial^2 V_i(\boldsymbol{0}, \hat{\boldsymbol{\alpha}}_i)}{\partial\boldsymbol{z}\partial\boldsymbol{z}^\mathsf{T}}\boldsymbol{z} + \frac{1}{2}\sum_{j,k,l\in[D]}\frac{\partial^3 V_i(\boldsymbol{0}, \tilde{\boldsymbol{\alpha}}_i)}{\partial z_j\partial z_k\partial\alpha_{il}}z_j z_k \Delta\alpha_{il} + \frac{1}{6}\sum_{j,k,l\in[D]}\frac{\partial^3 V_i(\tilde{\boldsymbol{z}}, \boldsymbol{\alpha}_i)}{\partial z_j\partial z_k\partial z_l}z_j z_k z_l,$$

for some $\tilde{\boldsymbol{z}}$ between $\boldsymbol{0}, \boldsymbol{z}$, some $\tilde{\boldsymbol{\alpha}}_i$ between $\hat{\boldsymbol{\alpha}}_i, \boldsymbol{\alpha}_i$. By the assumption that the moment generating function of $\boldsymbol{\eta}_i$ exists and by Lemma B.3, for the first term

$$\frac{1}{2T}\boldsymbol{z}^\mathsf{T}\frac{\partial^2 V_i(\boldsymbol{0}, \hat{\boldsymbol{\alpha}}_i)}{\partial\boldsymbol{z}\partial\boldsymbol{z}^\mathsf{T}}\boldsymbol{z} = \frac{1}{2T}\boldsymbol{z}^\mathsf{T}\int_0^T \frac{\boldsymbol{\eta}_i(t)\boldsymbol{\eta}_i(t)^\mathsf{T}}{\hat{\lambda}_i(t)}dt\boldsymbol{z} \to \frac{\boldsymbol{z}^\mathsf{T}I_i^*\boldsymbol{z}}{2}$$

with probability 1. For the second term, for any $j,k,l \in [D]$,

$$\left|\frac{1}{T}\frac{\partial^3 V_i(\boldsymbol{0}, \tilde{\boldsymbol{\alpha}}_i)}{\partial z_j\partial z_k\partial\alpha_{il}}\right| = \left|\frac{1}{T}\int_0^T \frac{\eta_{ij}(t)\eta_{ik}(t)\eta_{il}(t)}{\tilde{\lambda}_i^2(t)}dt\right| \leq \left|\frac{1}{T}\int_0^T \frac{\eta_{ij}(t)\eta_{ik}(t)\eta_{il}(t)}{\mu_i^2}dt\right|.$$

is uniformly bounded for any $\tilde{\boldsymbol{\alpha}}_i \geq 0$ with probability 1. For the third term, for any $i, j, k \in [D]$,

$$\left| \frac{1}{T} \frac{\partial^3 V_i(\tilde{\boldsymbol{z}}, \boldsymbol{\alpha}_i)}{\partial z_j \partial z_k \partial z_l} \right| = \left| \frac{1}{T} \int_0^T \frac{\eta_{ij}(t)\eta_{ik}(t)\eta_{il}(t)}{\lambda_i^2(t)} \exp\left( \lambda_i^{-1} \boldsymbol{\eta}_i^{\mathsf{T}}(t)\tilde{\boldsymbol{z}} \right) dt \right|$$

$$\leq \left| \frac{1}{T} \int_0^T \frac{\eta_{ij}(t)\eta_{ik}(t)\eta_{il}(t)}{\mu_i^2} \min\left\{ \exp\left( \mu_i^{-1} \boldsymbol{\eta}_i^{\mathsf{T}}(t)\tilde{\boldsymbol{z}} \right), 1 \right\} dt \right|,$$

is convex in $\boldsymbol{z}$. There exists a neighborhood $U$ of $\boldsymbol{0}$ such that the expectation of the term above for any $\boldsymbol{z} \in U$ is finite, and by its convexity, it is uniformly bounded in $U$ with probability 1.

Together, we have

$$\left| V_i(\boldsymbol{z}, \boldsymbol{\alpha}_i) - \frac{T}{2} \boldsymbol{z}^{\mathsf{T}} I_i^* \boldsymbol{z} \right| \leq o(T)\|\boldsymbol{z}\|^2 + O(T)\|\Delta\boldsymbol{\alpha}_i\|\|\boldsymbol{z}\|^2 + O(T)\|\boldsymbol{z}\|^3.$$

$\square$

*Proof of Proposition 1.* We prove a slightly weaker version: for any neighborhood $U_1$ of $\boldsymbol{\alpha}_i^*$, such that the diameter of $U_1$ is $o(1)$, the width of $\mathcal{C}_{i,\varepsilon} \cap U_1$ in $\alpha_{ij}$ converges to $2\sqrt{2\ln(K/\varepsilon)\sigma_{ij}^2/T}$ with probability 1.

Before proving the proposition, we explain why we choose $\boldsymbol{z}_1, \cdots, \boldsymbol{z}_K$ this way. Let $\boldsymbol{z}_1, \cdots, \boldsymbol{z}_{2D}$ be $\pm c_j I_i^{*-1} \boldsymbol{e}_j$, $c_j > 0$, $j = 1, \cdots, D$. By Lemma 3.1, we have

$$(\pm c_j I_i^{*-1} \boldsymbol{e}_j)^{\mathsf{T}} S_i(\boldsymbol{\alpha}) = (\pm c_j I_i^{*-1} \boldsymbol{e}_j)^{\mathsf{T}} T I_i^* (\boldsymbol{\alpha}_i - \hat{\boldsymbol{\alpha}}_i) + c_j \left( O(T)\|\boldsymbol{\alpha}_i - \hat{\boldsymbol{\alpha}}_i\|^2 + o(T)\|\boldsymbol{\alpha}_i - \hat{\boldsymbol{\alpha}}_i\| \right),$$

for any $\boldsymbol{\alpha}_i \geq 0$. Note that

$$(\pm c_j I_i^{*-1} \boldsymbol{e}_j)^{\mathsf{T}} T I_i^* (\boldsymbol{\alpha}_i - \hat{\boldsymbol{\alpha}}_i) = \pm c_j T(\alpha_{ij} - \hat{\alpha}_{ij}).$$

By (10), we have

$$V_i(\pm c_j I_i^{*-1} \boldsymbol{e}_j, \boldsymbol{\alpha}_i) = \frac{c^2 T}{2} \boldsymbol{e}_j^{\mathsf{T}} I_i^{*-1} \boldsymbol{e}_j + c^2 \left( o(T) + O(T)\|\boldsymbol{\alpha}_i - \hat{\boldsymbol{\alpha}}_i\| \right) + O(T)c^3.$$

The constraints

$$\boldsymbol{z}_k^{\mathsf{T}} S_i(\boldsymbol{\alpha}_i) - V_i(\boldsymbol{z}_k, \boldsymbol{\alpha}_i) \leq \ln(K/\varepsilon), \quad k = 1, \cdots, 2D$$

becomes

$$c_j T |\alpha_{ij} - \hat{\alpha}_{ij}| - \frac{c_j^2 T}{2} \sigma_{ij}^2 + o(T)c_j^2 + O(T)c_j^3 + (O(T)c_j^2 + o(T)c_j)\|\boldsymbol{\alpha}_i - \hat{\boldsymbol{\alpha}}_i\| + O(T)c_j\|\boldsymbol{\alpha}_i - \hat{\boldsymbol{\alpha}}_i\|^2$$
$$\leq \ln(K/\varepsilon), \quad j = 1, \cdots, D.$$

If all the $o(\cdot), O(\cdot)$ terms are negligible when $T \to \infty$, the width of $\mathcal{C}_{i,\varepsilon}$ in $\alpha_{ij}$ is

$$2\left( \frac{\ln(K/\varepsilon)}{c_j T} + \frac{c_j \sigma_{ij}^2}{2} \right),$$

and is minimized when

$$c_j = \sqrt{\frac{2\ln(K/\varepsilon)}{T\sigma_{ij}^2}}.$$

The $o(\cdot), O(\cdot)$ terms are indeed negligible with this choice of $c_j$, because the constraints now becomes

$$\sqrt{2T\ln(K/\varepsilon)/\sigma_{ij}^2}|\alpha_{ij} - \hat{\alpha}_{ij}| + o(T^{1/2})\|\boldsymbol{\alpha}_i - \hat{\boldsymbol{\alpha}}_i\| + O(T^{1/2})\|\boldsymbol{\alpha}_i - \hat{\boldsymbol{\alpha}}_i\|^2 \leq 2\ln(K/\varepsilon) + o(1), \quad (15)$$

$j = 1, \cdots, D$. Let $U_1$ be any neighborhood of $\boldsymbol{\alpha}_i^*$ with diameter $o(1)$. For any $\boldsymbol{\alpha}_i \in \mathcal{C}_{i,\varepsilon} \cap U_1$, we choose

$$j' = \underset{j \in [D]}{\arg\max} |\alpha_{ij} - \hat{\alpha}_{ij}|/\sigma_{ij}.$$

By the way we choose $j'$, $\|\boldsymbol{\alpha}_i - \hat{\boldsymbol{\alpha}}_i\|$ can be upper bounded by $|\alpha_{ij'} - \hat{\alpha}_{ij'}|$ up to some constant scale, and $|\alpha_{ij'} - \hat{\alpha}_{ij'}| = o(1)$. There is

$$\sqrt{2T\ln(K/\varepsilon)/\sigma_{ij'}^2}|\alpha_{ij'} - \hat{\alpha}_{ij'}| + o(T^{1/2})|\alpha_{ij'} - \hat{\alpha}_{ij'}| \leq 2\ln(K/\varepsilon) + o(1),$$

and

$$\frac{|\alpha_{ij'} - \hat{\alpha}_{ij'}|}{\sigma_{ij'}} \leq \sqrt{\frac{2\ln(K/\varepsilon)}{T}}(1 + o(1)).$$

Again by the way we choose $j'$, this inequality holds for any $j \in [D]$. So the width of $\mathcal{C}_{i,\varepsilon}$ in $\alpha_{ij}$ is upper bounded by $2\sqrt{2\ln(K/\varepsilon)\sigma_{ij}^2/T}(1 + o(1))$ with high probability. It is easy to see from (15) that there exists $\boldsymbol{\alpha}_i \in \mathcal{C}_{i,\varepsilon}$ with $\alpha_{ij} = \hat{\alpha}_{ij} \pm \sqrt{2\ln(K/\varepsilon)\sigma_{ij}^2/T}(1 - o(1))$. Together, we know that the width of $\mathcal{C}_{i,\varepsilon}$ converges to $2\sqrt{2\ln(K/\varepsilon)\sigma_{ij}^2/T}$ with probability 1. $\qquad\square$