[Reviews · NeurIPS 2020]

Review 1

Summary and Contributions: The authors develop a method to produce confidence sets for the MLE estimate in a Hawkes process over networks. The proposed method is non-asymptotic, in contrast to existing approaches, and achieves more accurate coverage in a semi-synthetic example.

Strengths: The problem of non-asymptotic inference for such models is an interesting one, and the proposed addresses this problem with theoretical results and one well-designed experiment.

Weaknesses: A key weakness of the general approach of pursing confidence intervals for parameters in a parametric model is the possibility for model mispecification. The authors perform inference for parameters in a parametric model, assuming the correctness of the model. The authors do perform a simulation where the data come from a distribution that is not exactly the Hawkes model, and demonstrate coverage of the "true parameters", defined as the parameters of a Hawke's model fit on a data set of much larger size. One possible shortcoming of the Hawke's model is that in full generality the number of parameters is quadratic in the number of nodes. As such, estimating this version of the model may be suitable only for a small number of nodes. This is not necessarily a problem with the current manuscript, but suggests that similar inferences for a simpler version of the model (for example, with a low-dimensional parameterization of the weights A) may be of interest in future work.

Correctness: The approach is reasonable and the claims are plausible.

Clarity: The is a very well-written paper. The main claims of the paper are clearly stated, and the flow of logic is clear. Small typo in line 227: "neuonal" should be "neuronal".

Relation to Prior Work: The related work section is accessible and effective, but the reviewer is not sufficiently familiar with this literature to know if important work has been missed.

Reproducibility: Yes

Additional Feedback: The reviewer thanks the authors for their clarification about the simulation setting used and finds the experiment with model mispeccification to be satisfactory. In addition, I agree with Reviewer 2 that since in practice the confidence set that is computed is an approximation to a non-asymptotic confidence set, the paper is currently cavalier in calling the method "non-asymptotic". This is acknowledged clearly in Section 3, but this should be more explicit in the abstract and introduction.


Review 2

Summary and Contributions: This paper deals with uncertainty quantification for multivariate Hawkes processes. An analytical method is introduced for constructing confidence intervals of the cross-productivity parameters of the network. These types of confidence bounds could be useful when estimating granger causality or solving problems in connectomics (which is the application the authors use to illustrate the method). To my knowledge this approach is novel and should be of interest to researchers working on multivariate Hawkes processes.

Strengths: The algorithm introduced seems fairly straightforward to implement, assuming the integrals in Alg. 1 can be solved or approximated. The authors also back up their method with theory, deriving several theorems to justify their approach. Furthermore, their approach is motivated by the need for UQ in smaller, non-asymptotic datasets. Given the interest in recent years in hawkes process networks and inferring neural connectivity, the paper should be of interest to the NeurIPS community.

Weaknesses: There is a pretty substantial flaw in the logic of the paper. In particular, the authors note on line 212-214 that their method requires g_k(\alpha_i^*), but they only have access to g_k(\hat{\alpha}_i). To get around this, they say here that when T\rightarrow \infty then one can use the first order Taylor approximation (line 214). However, the whole point of this method is to not have to let T go to infinity. And no analysis is given on the error introduced when making this approximation. Some details are also missing from the paper. How are the authors proposing to solve the integrals in section 3.3 that go into algorithm 1. Maybe in some limited cases these could be solved analytically, but in general I would assume you need some sort of quadrature in the algorithm. Another weakness is that UQ is only derived for the productivity. What about UQ for other parameters or the intensity itself? I think most statisticians would approach this model differently. A Bayesian approach would be employed which would give the full posterior. Such an approach would be model agnostic for the triggering kernel and could be estimated via HMC given the high-dimensionality. Why not use such an approach? What does this method give us in terms of computational cost? I also think the experiments are somewhat lacking. Given my concern about T not being large enough to use the first order taylor approximation, I would like to see at least some synthetic experiments that vary T and D and show when these CIs might be valid. Update: In the response the authors state that it is possible to perform numerical inversion of g_k. The authors also add an experiment in the response where they compare their CI estimate to the asymptotic estimate for varying T. I think this experiment is a good addition. However, it doesn't quite address whether their CI gives the right estimate, only that it is better than the asymptotic result. I think a comparison to either a sampling based CI (by repeatedly simulating hawkes and estimating parameters via MLE), or a Bayesian CI with uninformative priors, would be more useful.

Correctness: I think most of the claims seem correct. There is the issue noted above with this method actually being valid for small T.

Clarity: Yes, the paper is well written and easy to follow.

Relation to Prior Work: I think this paper should have been cited: Linderman, Scott W., Yixin Wang, and David M. Blei. "Bayesian inference for latent Hawkes processes." Advances in Neural Information Processing Systems (2017). It would have been nice to have a computational comparison of the present approach with Bayesian monte carlo.

Reproducibility: No

Additional Feedback:


Review 3

Summary and Contributions: This paper deals with parametric inference for a multivariate Hawkes process. More specifically, given data following a parametric Hawkes process (and hence, non iid data), the authors aim at providing non-asymptotic confidence sets with asymptotic control of their width, by using a martingale concentration inequality The authors first define a family of confidence bands, indexed by some vectors $z$, and they finally choose a value for those vectors $z$ that is based on an estimate of the Fisher information, and that guarantees that the asymptotic width of the final confidence band is only a constant times that of the asymptotic confidence set.

Strengths: The message of this paper is fairly simple and, I think, could be extended to a more general framework: From a realization of a parametric stationary point process, use a martingale inequality (Lemma B1) for a specific martingale (related to the so called "intrinsic variance", Lemma B2) in order to produce a confidence band. In my opinion, this work is interesting and relevant to the NeurIPS community, but suffers quite a poor structure. The two key lemmas B1 and B2 should definitely be in the main text, whereas Sections 2.1, 2.2 and 3.1 should be shortened down to a couple of paragraphs (asymptotic confidence bands are pretty standard and not new, and they are not really relevant in this work). Moreover, why only focus on Hawkes processes? It seems to me that the idea of this paper can be extended in far more generality.

Weaknesses: In my opinion, the main weakness of this work is that by focusing on these parametric Hawkes processes, the authors lose the reader and fail at sending the right message which, though, is very interesting, in my opinion. Moreover, some more effort should be dedicated to the writing, and especially the notation, which is a bit clumsy at certain times. One specific comment: Why not study the asymptotic width of your main confidence band (which should be the very heart of this paper!) in Corollary 1?

Correctness: The claims and their proofs seem correct to me and the empirical evaluation seems right. However, I could not find any proof (or any mention to a proof) for Theorem 3.3 and Corollary 1. If the authors consider that the proofs are straightforward, at least they should mention this and give a short explaination.

Clarity: I think that the paper needs to be restructured (see my comments above). Moreover, I have a few more specific comments about the writing. 1. Lines 88-89, the indices seem wrong to me. Under the sum, should it rather be "t in H_T" and in the sum, f(t) (without the index i)? 2. Line 121: I do not understand the last equality, where did the integral go? 3. Just before Theorem 3.2, write explicitly that you are going to define a family of confidence bands, indexed by z_1,...,z_K. It took me a while to understand why there suddenly were z's here. 4. Line 180: Remove "like". 5. Equation that follows Line 180: Did you rather mean \alpha_i^*, instead of \alpha_i? In the sequel, I think there are multiple other instances where the * seems to be missing (e.g., in Lemma 3.1). 6. Line 187: Specify whether Equation (10) holds uniformly in z or not. 7. In proposition 1, you should rephrase the conclusion. You write that as T goes to infinity, the width converges to something that still depends on T, which is mathematically incorrect. Either replace "converges to" with "behaves as", or rephrase the whole sentence. Same on Line 408. 8. Line 202: I do not understand what the departure set of the function z is. The notation is very unclear. Moreover, it seems that you treat (H_t)_t as a family of sets and as a filtration (Line 367), which is confusing. Please modify this and make the notation clear. 9. Line 215: By performing this approximation, you waste all your non-asymptotic analysis, since you lose the non-asymptotic coverage guarantee... 10. Line 227, replace "neuonal" with "neuronal" SUPPLEMENTARY MATERIAL: 11. Line 356-357: To me, "dN_t" is just a notation, but it does not stand for a mathematical object. Therefore, talking about its expectation or covariance makes no rigorous sense. Same in Line 171, the statement does not really make sense. 12. Line 365: Remove 's' from "results". 13. Line 378: In the computation, you are forgetting "(t)" (e.g., it should be \lambda(t) instead of just \lambda) 14. Line 378: This whole computation seems correct, but there is no rigorous justification (one should invoke dominated convergence in order to swap derivatives and conditional expectations). Please comment on that. 15.

Relation to Prior Work: The relation to prior work seems clearly discussed here.

Reproducibility: Yes

Additional Feedback:


Review 4

Summary and Contributions: The authors proposed an uncertainty quantification method for maximum likelihood estimate of network parameters in a multivariate hawkes process using concentration inequalities. Previous work has introduced a method based on classic asymptotic confidence intervals considering the MLE of Hawkes process to be asymptotically normal. The bound proposed in this method changes with time, hence can’t be bounded. Therefore, the authors have proposed a method for finding intrinsic variance which uses data to calculate the confidence interval. The confidence set proposed by authors is more general and in the form of convex polyhedron, thereby providing linear optimization. For the experiments, authors have shown the uncertainty quantification of connectivity between nodes in a neuronal network data and showed the effectiveness of the proposed approach.

Strengths: The paper is theoretically sound and the authors have provided relevant proofs and derivations for uncertainty quantification parameters in a HP using concentration inequalities. There has not been much recent work along this direction, hence it will revive uncertainty quantification of parameters in Hawkes process. The effectiveness of the proposed methodology is demonstrated on reconstructing neuronal network data.

Weaknesses: Some of the mathematical content and derivations could be moved to appendix and a more intuitive explanations would have been better. Readability of the paper is affected due to some derivations and proofs. The proposed work seems to be weak experimentally, and could have compared against related works on Bayesian modelling for hawkes process and on other data sets including more synthetic data sets and domains where hawkes process are used. A possible baseline for this work could be - Salehi, F., Trouleau, W., Grossglauser, M. and Thiran, P., 2019. Learning hawkes processes from a handful of events. In Advances in Neural Information Processing Systems (pp. 12715-12725). where authors have learnt posterior over parameters instead of point estimate and used it to estimate variance. Another possible baseline to compare against could be the bayesian hawkes models where a distribution is introduced on the prior and consequently an uncertainty estimation can be done. Lim, K.W., Lee, Y. and Ong, C.S., 2016, December. Bayesian Bivariate Hawkes. In Proceedings of the Workshop on Time Series Analytics and Applications (pp. 13-18). It will be better to mention real world applications of uncertainty modeling of the model parameters in a hawkes process in the introduction and broader impact section. For instance, proposed work could be useful for applications like community detection. Moreover, information about confidence intervals for influence parameters can be very useful for sensitive applications like crime forecasting and network detection in social media. A concrete example where uncertainty has been useful is ‘Model-based testing for space–time interaction using point processes. A related work along different direction is ‘ Quantifying uncertainty in a predictive model for popularity dynamics ‘ where the authors have tried to find uncertainty over predictions using a differential equation approach considering the branching process.

Correctness: The proposed methodology seems correct theoretically, and experiments conducted on neuronal data seems to support it. However the experiments are weak and not very convincing. A doubt is why the multivariate HP is trained on s smaller data set ended up learning a dense network. I would expect it to learn a even sparse network.

Clarity: Overall the structure of paper is good. It could be made more readable by providing more intuitive explanations, strong motivations and moving few mathematical content to appendix. The paper does not have a conclusion section.

Relation to Prior Work: The paper could have discussed related works on Bayesian modelling for hawkes process. Salehi, F., Trouleau, W., Grossglauser, M. and Thiran, P., 2019. Learning hawkes processes from a handful of events. In Advances in Neural Information Processing Systems (pp. 12715-12725). where authors have learnt posterior over parameters instead of point estimate. Their work has also shown variance over the parameters learnt. Another related work is the bayesian hawkes models where a distribution can be introduced on the prior and hence an uncertainty estimate can be done. Lim, K.W., Lee, Y. and Ong, C.S., 2016, December. Bayesian Bivariate Hawkes. In Proceedings of the Workshop on Time Series Analytics and Applications (pp. 13-18). Similar work has been introduced here. Sarma, S.V., Nguyen, D.P., Czanner, G., Wirth, S., Wilson, M.A., Suzuki, W. and Brown, E.N., 2011. Computing confidence intervals for point process models. Neural computation, 23(11), pp.2731-2745.

Reproducibility: Yes

Additional Feedback: ---------------------------AFTER REBUTTAL----------------------------- I thank the authors for the rebuttal and acknowledging some of the concerns. Taking into account the other reviewers' comments and the author's feedback, I maintain my original evaluation.


Review 5

Summary and Contributions: I was called in as a last-minute emergency reviewer for this paper. This is a very sophisticated paper that solves a hard and interesting problem---estimation of Hawkes processes is reasonably well-studied, but uncertainty quantification is much less understood. This paper in particular looks at the network multivariate Hawkes process MLE, and provides nontrivial martingale-based confidence sets. Theorem 3.2 and 3.3 are based on a new continuous-time martingale, and the intrinsic variance techniques are quite contemporary. The application to learning neuronal networks based on spike train data is also fascinating. The paper seems ready for publication, the methodology and theory and application all look fairly well developed.

Strengths: N/A---Emergency Review

Weaknesses: N/A---Emergency Review

Correctness: N/A---Emergency Review

Clarity: N/A---Emergency Review

Relation to Prior Work: N/A---Emergency Review

Reproducibility: Yes

Additional Feedback: N/A---Emergency Review

[Author Response · NeurIPS 2020]

We greatly thank the reviewers for their constructive comments. We would like to specifically emphasize that the
proposed frequentist confidence interval method is different but complementary to a Bayesian UQ approach; please see
our response to Reviewer 2. We will also include the suggested references for Bayesian UQ methods.

**Response to Reviewer 1**. Thank you for the valuable feedback about model misspecification and low-rank parameteri-
zation. In the neuronal connectivity application, while it is commonly agreed that Hawkes process is a good fit for the
spike train data, such data are generated from a scientific model (i.e., **not** exactly Hawkes; in fact, the synthetic data are
generated by solving a complex PDE system), so there is indeed a slight model mismatch in our numerical example. In
future work, it will be very interesting to study the influence of model mismatch and the confidence sequences when the
weight matrix $A$ is low-rank or sparse.

**Response to Reviewer 2**. 1) The first-order Taylor approximation of $g_k(\alpha_i^*)$ at $\widehat{\alpha}_i$ is for the purpose of giving a
tangible form (polyhedron) of the confidence set. However, we can definitely evaluate $g_k$ at every $\alpha_i$ and perform
numerical inversion to find the confidence set. Moreover, we have empirically validated the performance of the
confidence set from Algorithm 1 in the numerical example. 2) The integrals in Section 3.3 (which go into Algorithm 1)
is a one-dimensional integral over $t$, and hence can be solved efficiently by standard quadrature techniques. 3) The UQ
for the intensity can be derived from the UQ for the weights $A$; this will be an interesting future work.
4) **Comment on the comparison with Bayesian approach.** Given sufficient computing resources and little domain
knowledge, we agree that a fully Bayesian nonparametric approach may provide a richer quantification of uncertainty.
However, in certain applications, there is scientific evidence for a *parametric* form of the triggering kernel (e.g., [Beggs
(2008)] for our neural spike data application). For large networks, the full posterior may also be computationally
expensive to sample with HMC, since one needs to tune both the stepsize and number of leapfrog steps, among other
settings. We will add further context on when our method may be more or less preferable to a fully Bayesian approach
in practice.
5) Due to space limitation, we did not include in the paper a simulated example of confidence bands as a function of $T$.
We show here a small example with 5 nodes in Fig. 1. We see that the proposed CIs are valid and becomes narrower as
$T$ increases, as desired. Moreover, when $T$ is small, Fig. 1 also shows that the asymptotic CI can have poor coverage
performance (i.e., it does not contain the true parameter), whereas our proposed method uniformly covers the true
parameter even for small $T$. We have similar observations for recovering other edges in this example.

28
**Response to Reviewer 3**. We agree that the idea can be extended beyond Hawkes
processes, and it would be an interesting future direction.
1) The reason for focusing on the Hawkes model is motivated by the neural
connectivity application, and we would like to generalize this to broader models
in future work. Moreover, we will adjust Section 2-3 as suggested.
2) The width of the main confidence band in the non-asymptotic case in Corol-
lary 1 will be similar to that of the asymptotic confidence band width shown in
Proposition 1, since the estimated Fisher information $\hat{I}_i$ will converge to the true
$I_i^*$ as $T \to \infty$.
3) Proof of Theorem 3.3 mirrors the proof of Theorem 3.2 by replacing the fixed
vector $z$ with the adapted measurable function $z : (\mathcal{H}_t)_{t=0}^T \to \mathbb{R}^D$, and Corollary
1 is a straightforward consequence of Theorem 3.3 by considering multiple $z$
functions, the choice of $z$ in Corollary 1 is inspired by Proposition 1.
4) An example of the integral in Line 121 for the exponential kernel function
is given in Appendix A; The $\alpha_i$ in Lemma 3.1 actually can be an arbitrary $\alpha_i$,
not necessarily $\alpha_i^*$; Equation (10) holds uniformly in $z$; the departure set of
the function $z$ means that $z$ is determined by past event times; as justified in
Response to Reviewer 2 - point 1, the approximation aims to provide a tangible
form of confidence set and is validated empirically, and we can also evaluate $g_k$
numerically to find the exact confidence set; the analysis on $dN_t$ is a standard
technique in Hawkes literature and is commonly seen in related literatures, such
as [Hawkes (1971)] [Bacry and Muzy (2016)].
5) Comment on Line 378: We appreciate reviewer's careful reading and sug-
gestion; we believe the final conclusion is correct (as validated by numerical
examples); we will make this more rigorous in the full paper.

Figure 1: The CIs for two select
edges at level $\varepsilon = 0.05$ over time
$T$, for a Hawkes network with 5
nodes and influence functions are
$\varphi_{ij}(t) = e^{-t}$. In this picture, the
solid blackline is true $\alpha_{ij}$ and the
dashed line is the zero level. In
each picture, the two blue curves
outline the proposed CIs, and the
two red curves outline the asymp-
totic CI. Note that the proposed CI
uniformly covers the true parameter
even when $T$ is small, whereas the
asymptotic CI can have poor cover-
age when $T$ is small. This shows
the advantage of our method over
the asymptotic approach, and that
our method works well for small $T$.

**Response to Reviewer 4**.
1) In the revised paper, we will add a paragraph to discuss the difference between our parametric approach with Bayesian
model and add all mentioned references. 2) Currently there is no sparsity structure imposed in the maximum likelihood
estimate, since our main goal is to obtain confidence intervals rather than point estimators; this is an interesting direction
and we will leave it for future work.

[Meta-Review · NeurIPS 2020]

This paper is makes sophisticated contributions on the theoretical and methodological fronts along with an interesting practical example. These network process models are fairly difficult to handle probabilistically and the continuous-time methods developed here are quite rigorous. The reviewers raised some concerns about comparisons to Bayesian nonparametric approaches (esp. R2) but I agree with the authors response that this paper develops a fairly different frequentist framework that is complementary to existing approaches in both their theoretical validity and computational efficiency. However, I do concur with R2 that some simulations should be added that verify coverage for small T, but this does not require a resubmission and can be added for the final version. All reviewers agreed that this work is novel and would be of interest to researchers employing Hawkes processes, and in my judgment R2 was slightly harsh with the final score. I recommend the authors to take the couple of suggestions into account (eg: experiments for small T), in order to strengthen an already very nice paper.